# Dynamical coupling between a nuclear spin ensemble and electromechanical phonons

Yuma Okazaki [1,2], Imran Mahboob[1], Koji Onomitsu[1], Satoshi Sasaki[1], Shuji Nakamura[2], Nobu-Hisa Kaneko [2] & Hiroshi Yamaguchi[1]

Dynamical coupling with high-quality factor resonators is essential in a wide variety of hybrid quantum systems such as circuit quantum electrodynamics and opto/electromechanical systems. Nuclear spins in solids have a long relaxation time and thus have the potential to be implemented into quantum memories and sensors. However, state manipulation of nuclear spins requires high-magnetic fields, which is incompatible with state-of-the-art quantum hybrid systems based on superconducting microwave resonators. Here we investigate an electromechanical resonator whose electrically tunable phonon state imparts a dynamically oscillating strain field to the nuclear spin ensemble located within it. As a consequence of the dynamical strain, we observe both nuclear magnetic resonance (NMR) frequency shifts and NMR sidebands generated by the electromechanical phonons. This prototype system potentially opens up quantum state engineering for nuclear spins, such as coherent coupling between sound and nuclei, and mechanical cooling of solid-state nuclei.

[1] NTT Basic Research Laboratories, NTT Corporation, 3-1 Morinosato-Wakamiya, Atsugi, Kanagawa 243-0198, Japan. [2] National Metrology Institute of Japan (NMIJ), National Institute of Advanced Industrial Science and Technology (AIST), Tsukuba 305-8563, Japan. Correspondence and requests for materials should be addressed to Y.O. (email: yuma.okazaki@aist.go.jp) or to H.Y. (email: yamaguchi.hiroshi@lab.ntt.co.jp)

D ynamical coupling to a high-quality factor (high-Q) resonator, i.e., an interaction with a rapidly oscillating degree of freedom in a resonator, is essential to implementing a wide variety of quantum hybrid devices. For instance, in circuit quantum electrodynamics (QED), which implements dynamical coupling between a superconducting microwave resonator and an artificial atom; quantum non-demolition measurements of the two-level system via the resonator's frequency shifts have been developed[1,2]. On the other hand the parametric coupling between photons and phonons (phonons and phonons) in opto-mechanics[3] (in coupled mechanical systems) have been used to measure the position of the mechanical resonator below the standard quantum limit[4]. The resultant sidebands generated on the photonic (phononic) resonance have also been harnessed for sideband cooling the thermal phonons down to the quantum ground state when pumping the red sideband[5–7], and to generate an inter-resonator quantum entanglement[8–12].

Nuclear spins in solids have long relaxation times because of their weak interaction with the surrounding environment, and thus are expected to be utilized for various applications such as quantum information processing[13–15] and quantum magnetic sensors[16]. Consequently incorporating the abovementioned concept of coupling a resonator to nuclear spins has the potential to open another class of manipulation and detection methods for the nuclear spins. The concept of circuit QED has already been implemented with electron spins using a superconducting microwave resonator[17–20]. In contrast, a similar approach based on superconducting resonator (necessary for their high-Q) is difficult for nuclear spins, because the nuclear magnetic resonance (NMR) used to detect and manipulate their states requires high-magnetic fields of a few Tesla. To address this, a natural alternative would be to couple the nuclei to an electromechanical resonator that is compatible with high-magnetic fields, but despite this promise, an experimental demonstration of such a hybrid system remains unexplored.

In the first step towards such a resonator-based hybrid system with a nuclear spin ensemble, we have developed an electromechanical system that imparts dynamical strain to nuclear spins naturally located within it. In this realization, the nuclear spins interact with the fundamental mode of the resonator at 1.7 MHz via mechanical strain in the quadrupole interaction[21,22]. The NMR of the local nuclei in the resonator can be selectively detected by resistively detected NMR method using the integer quantum Hall effect through a nano-fabricated transport channel at the clamping point of the resonator[15]. The quadrupole interaction of nuclear spins in solids is important and has a long history in standard NMR and nuclear acoustic resonance experiments[23]. We note however that most of the work reported so far has focused on static strain[21,22,24–37] or weak oscillating strain induced by ultrasound applied to bulk specimens[38]. In contrast, here we achieve larger strain regime of $10^{-3}$ at 1.7 MHz with the aid of a high-Q mechanical resonator that can enhance the mechanical displacement (which induces strain) by a factor of $Q$. As a consequence of this resonator-enhanced dynamical strain effect, we observe both red and blue sidebands on the NMR which correspond to nuclear spin transitions assisted by the absorption and emission of electromechanical phonons. Moreover, frequency shifts in the NMR are also observed, which scaled with the strain from the mechanical motion thus enabling the nuclear spin states to be mechanically manipulated.

## Results

### Electromechanical resonator and resistively detected NMR. A 50-μm-long doubly clamped electromechanical resonator shown in Fig. 1a was fabricated from a GaAs/AlGaAs heterowafer using

sacrificial layer etching (see Methods and Supplementary Fig. 1). Figure 1b details the motional piezovoltage $V_d$ measured at different actuation voltages $V_a$, showing a resonance peak which corresponds to the fundamental flexural mode[39,40] with frequency $\omega_M/2\pi = 1.716$ MHz and quality factor $Q = 5 \times 10^4$. With increasing $V_a$, the mechanical amplitude is deformed into a sawtooth-shape reflecting the emergence of a nonlinearity in the underlying harmonic potential[41]. This nonlinear feature is further enhanced in the strong driving regime, where the normal and shear strains can become as high as $\varepsilon_{zz} \sim 1 \times 10^{-3}$ and $\varepsilon_{xy} \sim 2 \times 10^{-3}$, respectively (see below and Supplementary Note 1).

This GaAs-based resonator hosts three isotopes $^{75}$As, $^{69}$Ga, and $^{71}$Ga, all of which are quadrupole nuclei having spin $I = 3/2$. Among them $^{75}$As has the largest quadrupole moment and the strongest coupling to strain and therefore the effects generated from electromechanical strain are most favorable for this element. The fundamental flexural mode of this resonator creates strain that is magnified at the clamping points (see Supplementary Fig. 2) and thus detection of NMR for the strain-modified nuclei is made at this location. To this end, a resistively detected NMR method is adopted that uses breakdown of the integer quantum Hall effect (see Methods). Figure 1d shows the NMR measured from the $^{75}$As nuclei at the clamping point, which is unstrained as the mechanical vibration is not activated and the peak frequency $\omega_0$ measured at different external field $B_0$ shows a linear dependence (see Fig. 1e). The corresponding gyromagnetic ratio $\gamma/2\pi = 7.3$ MHz T$^{-1}$ extracted from a linear fit confirms that the observed peak originates from the NMR of $^{75}$As nuclei.

**Sideband NMR assisted by electromechanical phonons.** When the electromechanical resonator is driven, it can periodically modulate the quadrupole term associated with the nuclear spins via the motion induced strain. As a consequence of this off-resonant parametric modulation, nuclear spins can form phonon-dressed states, which emerge as sideband NMR peaks as schematically depicted in Fig. 2a. A red (blue) transition can be induced with the absorption (emission) of phonons from (into) the mechanics via the quadrupole interaction and hence the additional NMR peak appears at $\omega_0 - (+)\omega_M$. Figure 2b–d (2e) show the observation (simulation) of red and blue sideband NMR peaks for the nuclei under the driven electromechanical oscillation with $V_a = 40$ mV (point 5 in Fig. 3a). The strongest sideband NMR peaks are observed with $^{75}$As whilst the weakest with $^{71}$Ga, which reflects the magnitude of the quadrupole interaction between the nuclear spins and mechanical strain. We note that the apparently narrower NMR peak of $^{71}$Ga is due to the limited signal-to-noise ratio for this nuclear species, which is relatively scarce and has a weaker quadrupole interaction.

To better understand the observed sideband NMR peaks, the spectra were numerically simulated using a model Hamiltonian that included an oscillating quadrupole interaction associated with the driven mechanical motion:

$$\hat{H} = \hat{H}_Z + \hat{H}_Q(t) + \cos(\omega_{RF}t)\hat{H}_{RF}. \tag{1}$$

Here the first term describes the Zeeman effect, $\hat{H}_Z = -\hbar\omega_0\hat{I}_z$, where $\omega_0/2\pi = \gamma B_0$ with $\gamma$ being the gyromagnetic ratio of the relevant nucleus, $\hat{I}_z$ is the z-component of the Pauli spin operator $\hat{I}$, and $\hbar$ the reduced Planck constant. For a nucleus with spin $I = 3/2$, four spin states $|m\rangle$ are Zeeman split to form equally spaced energy levels separated by $\hbar\omega_0$. The effect of strain-mediated dynamical coupling from a driven mechanical resonator on nuclear spins is captured by the dynamic quadrupole interaction described by the second term $\hat{H}_Q(t) = x(t)\hat{J}_Q$, which is proportional to the time-dependent mechanical displacement $x(t)$. Here

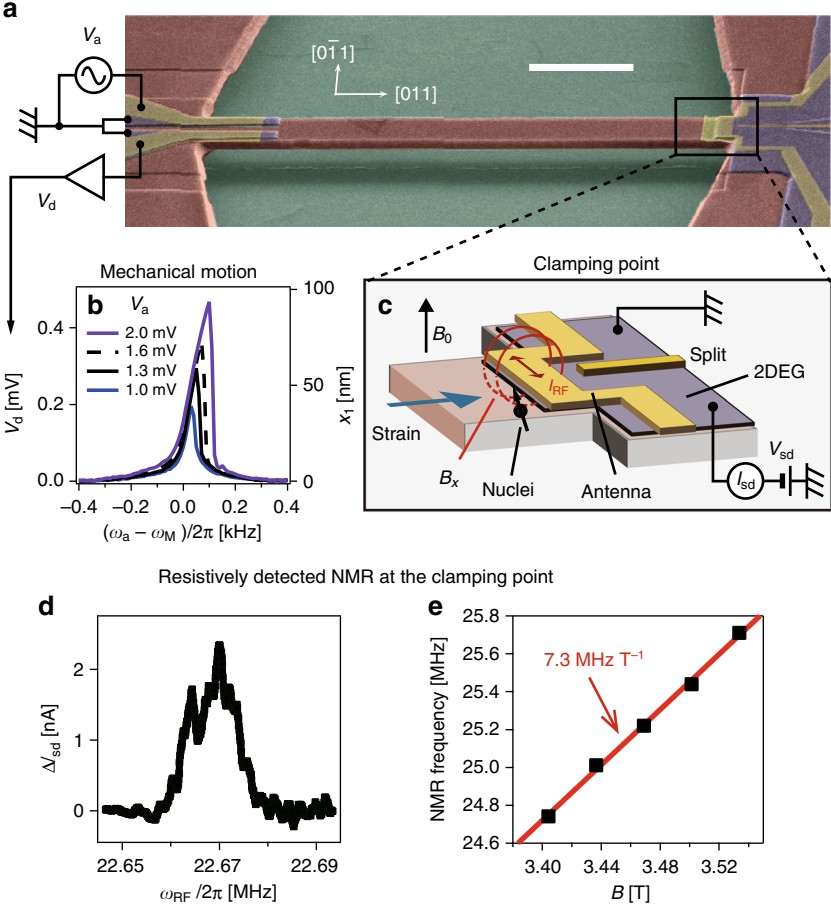

**Fig. 1** Experimental setup. **a** False-color scanning electron microscope image of the device (scale bar, 10 µm). A GaAs-based doubly clamped electromechanical resonator is suspended from the substrate. Two Schottky gate electrodes are defined on the left clamping point in order to piezoelectrically actuate and detect the mechanical motion. **b** Frequency response of the mechanical resonator with different actuation voltage $V_a$ plotted as a function of actuation frequency $\omega_a/2\pi$ with $\omega_M/2\pi = 1.716$ MHz. The right axis shows the calibrated mechanical amplitude $x_1$ at the center of the beam. **c** A schematic of the device structure at the clamping point, where the maximum strain can be induced from the fundamental flexural mode of this resonator. A static magnetic field $B_0$ is applied perpendicular to the 2DEG, while a transverse magnetic field $B_x$ is irradiated to the nuclei underneath the antena gate to which an RF current $I_{RF}$ with a frequency $\omega_{RF}$ is also applied. **d** A plot of $\Delta I_{sd}$ ($\equiv I_{sd} - I_0$ with $I_0$ being the offset current) at $B_0 = 3.1$ T measured as a function of $\omega_{RF}$ without mechanical actuation. **e** A plot of the NMR peak frequency as a function of $B_0$, showing linear $B_0$-dependence, from which a gyromagnetic ratio of 7.3 MHz T$^{-1}$ for As$^{75}$ is determined

the quadrupole operator $\hat{J}_Q$ is

$$\hat{J}_Q = A_1\left[3\hat{I}_z^2 - I(I+1)\right]/3 + iA_2\left[\hat{I}_+^2 - \hat{I}_-^2\right], \quad (2)$$

where $A_1$ and $A_2$ are the quadrupole coupling constants at the clamping point and $\hat{I}_+$ and $\hat{I}_-$ are the spin ladder operators (see Supplementary Note 1). The NMR transition triggered by absorbing a photon $\omega_{RF}$ irradiated by the antenna gate is described by the third term with $\hat{H}_{RF} = -\gamma\hbar B_x\hat{I}_x$, where $\hat{I}_x$ is the $x$-component of the Pauli spin operator and $B_x$ is the corresponding transverse magnetic field. Equation (1) can simulate the magnetization $\langle\hat{I}_z\rangle$ driven by RF irradiation and its amplitude reflects the NMR response of the nuclei under the presence of an oscillating strain field.

Figure 2e shows the numerical simulation extracted from this framework within the rotating-wave approximation of both the main and sideband NMR spectra associated with $^{69}$Ga nuclei (see Methods). The simulation not only reproduces the sideband NMR peaks but also the ratio of the signal amplitude between the main peak and the sidebands as observed in the experiments. However the simulated sidebands are narrower than the main NMR peak and this feature is not seen in the experimental

spectrum as this is acquired from an ensemble average (in contrast to the simulation) that is inhomogeneously broadened by the spatial distribution of the conduction electrons and the strain. This agreement with the theory confirms that the observed sideband peaks in Fig. 2b–d are indeed associated with nuclear spin transitions aided by electromechanical phonons via the quadrupole coupling.

**Dispersive shifts induced by a driven mechanical resonator.** From the overall similarity between our system and a circuit QED device embedded with an electromechanical resonator[42], one can predict frequency shifts in the NMR caused by a driven mechanical resonator. This feature resembles the ac-Stark shift in a circuit QED system[2,43] but now the driven mechanical motion that is off-resonant with the qubit transition frequency is treated as Floquet perturbation resulting in an energy shift, that was dubbed as the mechanical/phonon ac-Stark shift[42].

In order to investigate this effect in the present system, the NMR spectra of $^{75}$As nuclei were measured under the influence of intense AC electromechanical strain as shown in Fig. 3b. The bottom spectrum shows the NMR at zero mechanical

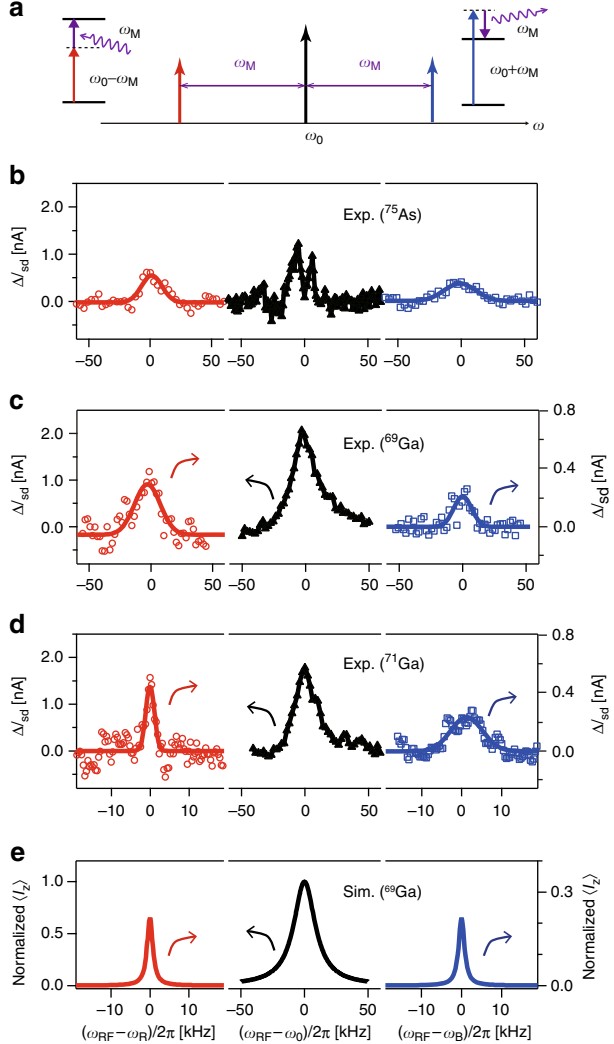

**Fig. 2** NMR sidebands induced by electromechanical phonons. **a** A red (blue) sideband NMR peak can be induced with the absorption (emission) of an electromechanical phonons. **b**–**d** The main and red/blue sideband NMR spectra with relevant isotopes, $^{75}$As, $^{69}$Ga, and $^{71}$Ga from top to bottom, respectively. Each panel shows $\Delta I_{sd}$ as function of $\omega_{RF}$ with $\omega_{R(B)} = \omega_0-(+)\omega_M$, where $\omega_0(^{75}\text{As})/2\pi = 22.63$ MHz, $\omega_0(^{69}\text{Ga})/2\pi = 31.79$ MHz, and $\omega_0(^{71}\text{Ga})/2\pi = 40.39$ MHz at $B_0 = 3.1$ T when the resonator is driven with $V_a = 40$ mV at point 5 in Fig. 3a. The solid lines in the sideband NMR spectra show Gaussian fits of the data. **e** The simulated NMR spectra for $^{69}$Ga using the rotating-wave approximation, which are normalized by the amplitude of the main NMR peak (see Methods)

displacement with a doublet peak structure as indicated by the solid circle (main peak) and open square (sub peak). However this figure also reveals that as the mechanical displacement is increased the main peak undergoes a blue shift whereas the sub peak remains fixed. To elucidate the underlaying physics of these observations, the NMR spectra are simulated using the same model Hamiltonian in Eq. (1) that describes nuclear spins under the influence of a rapidly oscillating quadrupole interaction arising from the driven electromechanical motion (see Methods) and are shown in Fig. 3c. The simulated NMR spectra (generated using the same parameters as in Fig. 2e but now with a variable displacement $x(t)$) reproduce the observed frequency shift, i.e., blue shift of the main peak (solid circle) and no shift of the sub peak (open square), again confirming that the experimental

observations can be accommodated by the oscillating quadrupole Hamiltonian. To further quantify the simulated spectra, the experimental NMR frequencies of the main- and sub-peaks extracted from the measured NMR spectra are collated as a function of the physical displacement of the fundamental flexural mode $x_1$. The resultant plot in Fig. 3d shows that the experimental frequency shift agrees with the simulated frequency shift.

In order to decipher the mechanisms underlying the observed frequency shift, the model Hamiltonian is further analyzed. In the absence of a quadrupole interaction between nuclei and mechanics, the bare nuclei with $I = 3/2$ shows degenerate Zeemann split states that are equally spaced by $\hbar\omega_0$ as shown in Fig. 4. Next the quadrupole interaction is treated using Floquet perturbation theory (see Methods), which notably indicates that the dynamic (oscillating) component of the perturbation term has no contribution at the first order while it has a non-zero value at higher order. Consequently the first-order energy shift $\delta$ can simply be induced by static mechanical strain. The analytical expression of $\delta$ (see Methods for the derivation) indicates that its sign is positive (negative) for $|m = \pm 1/2\rangle$ ($|\pm 3/2\rangle$), which in turn lifts degeneracy as depicted in Fig. 4 and is experimentally observed as split NMR peaks even in the absence of ac strain in the lowest plot in Fig. 3b. Indeed, such NMR peak structure is commonly observed in GaAs heterostructures[15] and is due to residual static strain that is incorporated during crystal growth and nano fabrication. Similarly, the sign of the second-order energy shift $\chi$ is positive (negative) for $|-1/2\rangle$ and $|-3/2\rangle$ ($|1/2\rangle$ and $|3/2\rangle$) and the dynamical electromechanical strain only blue shifts the resonant transition $|1/2\rangle \leftrightarrow -|1/2\rangle$, whereas the other transitions remain unchanged as shown in Fig. 4. This is in contrast to the case with static strain where the transition $|3/2\rangle \leftrightarrow |1/2\rangle$ ($|-3/2\rangle \leftrightarrow |-1/2\rangle$) is blue (red) shifted while the $|1/2\rangle \leftrightarrow |-1/2\rangle$ transition remains unchanged. Consequently in the NMR spectra shown in Fig. 3b, d, the blue shifting main peak corresponds to the transition $|1/2\rangle \leftrightarrow |-1/2\rangle$ stemming from non-adiabatic dynamical strain originating from electromechanical phonons, whilst the sub peak (open square) corresponds to the $|-3/2\rangle \leftrightarrow |-1/2\rangle$ transition which arises from the residual static strain given by $-2\delta$.

## Discussion

The NMR spectra of $^{75}$As, $^{69}$Ga, and $^{71}$Ga nuclei under the influence of intense periodic lattice strain induced by the large displacements of a strongly-driven doubly clamped electromechanical resonator were investigated. The resultant resistively detected NMR showed sideband peaks, which indicated the parametric coupling between nuclei and mechanics with the nuclear spin transitions assisted by electromechanical phonons. In addition, NMR frequency shifts were also observed that stemmed from non-adiabatic dynamics that scaled with the mechanical displacements. The agreement with theoretical modeling in both experiments confirms that our system faithfully implements dynamical coupling between a nuclear spin ensemble and an electromechanical resonator via mechanical strain.

In the present system, frequency shifts in the mechanical resonance from the NMR have yet to be observed, which indicates that the strain-mediated coupling observed in this system corresponds to a weak coupling regime. For more advanced dynamic effects such as coherent state transfer between a nuclear spin ensemble and electromechanical phonons and electromechanical detection of the NMR requires stronger coupling between nuclei and mechanics. In case of an ensemble, the effective coupling can be scaled with $\sqrt{N_P N_{NS}}$ due to collective effects[44,45], where $N_{NS}$ is the number of nuclear spins equally

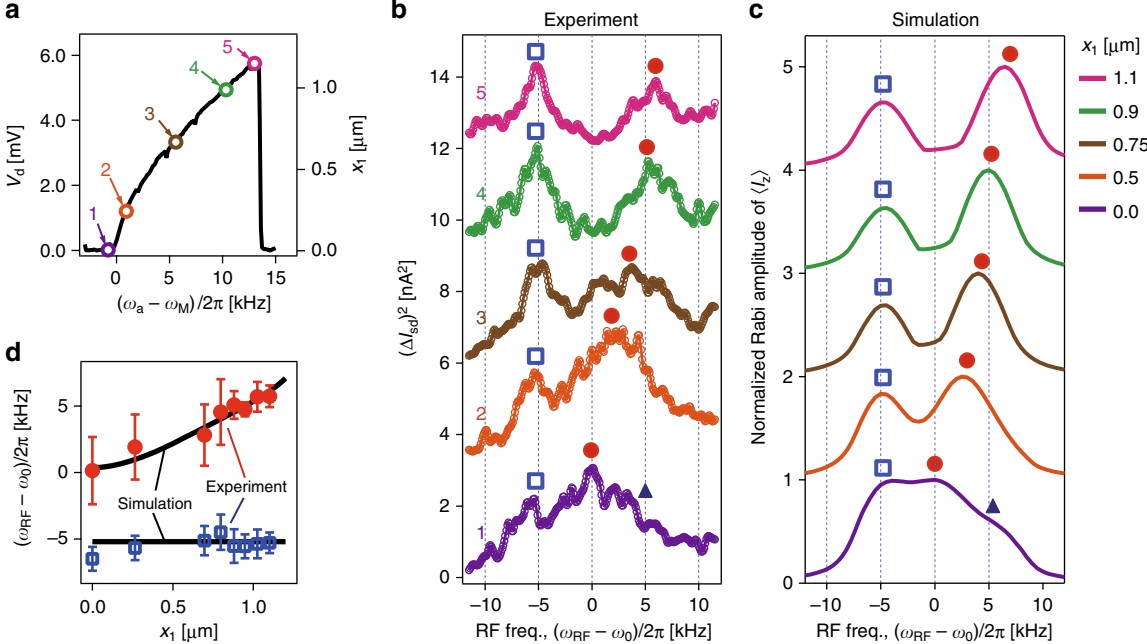

**Fig. 3** NMR frequency shift induced by a driven electromechanical strain. **a** A plot of the mechanical amplitude $x_1$ with stronger actuation: $V_a = 40$ mV. **b** NMR spectra for $^{75}$As nuclei at $B_0 = 3.1$ T under intense mechanical oscillations as a function of $\omega_{RF}$ with $\omega_0/2\pi = 22.606$ MHz. From bottom to top, the spectra are measured at $(\omega_a - \omega_M)/2\pi = -0.3, 1.0, 6.3, 10.8,$ and $13.0$ kHz with a fixed actuation $V_a = 40$ mV as indicated by the colored open circles in **a**. **c** The numerical simulation of the NMR spectra which shows the normalized oscillation amplitude of $\langle I_z \rangle$ for $x_1 = 0, 0.5, 0.75, 0.9, 1.1$ μm from bottom to top, respectively. **d** A plot of the measured NMR peak frequency as a function of $x_1$, where the open square (solid circle) symbol corresponds to the left (center) peak in **b** determined from the Lorenz fit to the spectra. The error bars correspond to the peak width determined from the fitting. The solid lines show the theoretical frequency shift derived from the numerical simulation

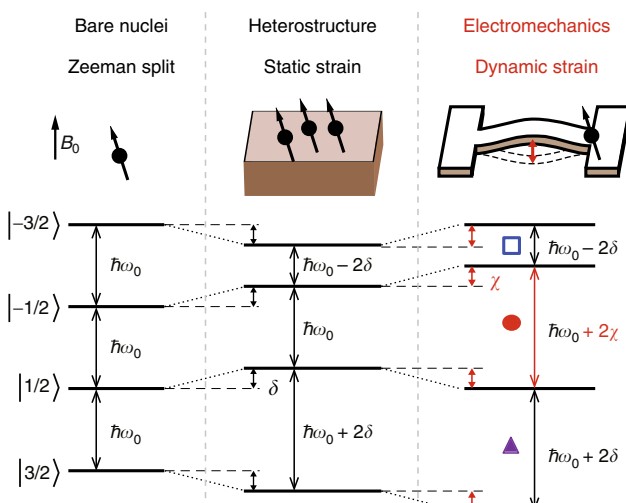

**Fig. 4** Energy diagram of the nuclei under electromechanical strain. The energy level diagram of the nuclear spin $I = 3/2$ under the influence of static and dynamic mechanical strain. $\delta$ ($\chi$) represents the energy level shift induced by static (dynamic) mechanical strain. The transitions labeled with the open square and solid circle correspond to the features in the NMR spectra in Fig. 3b, c with the same markings. The transition given by the solid triangle, although marked in Fig. 3b, c, is actually not observed in both the experiment and the simulation due to it being masked by the blue shifting main peak

coupled to the mechanical resonator which contains $N_P$ phonons. Indeed such collective enhancement of coupling enables quantum entanglement to be generated between an ensemble and a resonator as demonstrated with an electron spin ensemble[17,18,46]. To

address the future possibility of accessing the strong-coupling regime in this platform, the coupling rate between a single $^{75}$As nucleus and the zero-point motion of the resonator is estimated to be $g_0 \sim A_1 x_{ZP} = 0.15$ mHz, where $x_{zp} = 3$ fm is the amplitude of the zero-point fluctuations in the present mechanical resonator[40]. The strong-coupling regime requires $\sqrt{N_P N_{NS}} g_0 > \max(\Delta f_M, \Delta f_{NS})$, where $\Delta f_M$ ($\Delta f_{NS}$) is the frequency bandwidth of the mechanical resonance (NMR peak). In the present work, $\Delta f_M = 25$ Hz and $\Delta f_{NS} = 5$ kHz, where the NMR spectrum is inhomogeneously broadened. However $N \sim 10^{10}$ nuclear spins can be obtained in a volume of $\sim(100$ nm$)^3$, which is much smaller than the electromechanical resonator, in which case the strong-coupling regime is achievable with $N_P \sim 10^4$ phonons, where these requirements would be accessible by simply optimizing the design of the resonator.

Further enhancement of this parametric coupling is necessary to achieving quantum coherent coupling, i.e., strong-coupling with $N_P < 1$. This objective could be attained by using strain-engineered soft-clamped ultrahigh-Q mechanical resonators[47,48], which can have a narrow linewidth down to mHz, and by improving the crystal quality to suppress the inhomogeneous broadening of the NMR spectrum as well as using quantum dots for highly polarized nuclear spin states[49]. On the other hand, a resonator with the frequency greater than the NMR frequency would be required for sideband cooling[3–7,50] of the nuclear spins which would necessitate changing the material of the resonator and/or making it shorter and lighter where both could be achieved with semiconductor nanowires[51,52]. For future application to a quantum coherent hybrid system, the decoherence induced by the unpolarized nuclear spins, which are coupled to mechanical motion at the clamping points (but unpolarized by the electron transport) needs to be suppressed. An optimized device structure where the mechanically strained clamping points

completely overlap with the nuclei polarized from the electron transport can in principle achieve this objective. Ultimately optical access to the sample will enable the nuclear bath in the entire resonator to be polarized[22,53], thus enabling quantum coherent coupling between nuclei and phonons. Our observation of dynamical coupling between an nuclear spin ensemble and electromechanical phonons combined with these improvements is a prototype for electromechanical resonator-nuclear spin hybrid systems.

## Methods

**Sample fabrication.** A doubly clamped electromechanical resonator with a length of 50 μm, width of 6 μm, and a thickness of 1 μm was fabricated from high-quality modulation-doped GaAs/Al$_{0.3}$Ga$_{0.7}$As heterowafer grown by molecular beam epitaxy on an undoped GaAs [100] substrate[39,40]. The beam resonator was defined along the [011] crystal axis. A single heterojunction located 90 nm below the surface sustains a two-dimensional electron gas (2DEG) with density $n = 2.7 \times 10^{11}$ cm$^{-2}$. The main heterolayer that constitutes the beam resonator was grown on an aluminum-rich Al$_{0.65}$Ga$_{0.35}$As sacrificial layer to enable it to be suspended from the substrate. The fine mesa structure on the clamping points was defined by electron-beam lithography and wet-etching. The antenna gate and the Schottky electrodes formed thereon were fabricated via electron-beam lithography and deposited with 1 nm/25 nm of Cr/Au. The final step to suspend the beam resonator was defined by photo lithography and wet-etched to expose the sacrificial layer. The sacrificial layer was then selectively removed with dilute hydrogen fluoride (10 wt%).

**Measurement setup.** Supplementary Figure 1a shows the experimental setup. All measurements were performed in a high-vacuum insert (<10$^{-5}$ Pa) in a dilution refrigerator with a base temperature (<80 mK). Source-drain current $I_{sd}$ is measured in a two-terminal configuration using a source measure unit with a bias voltage $V_{sd}$. In order to irradiate the RF transverse magnetic field in the NMR measurements, a signal generator is connected to one end of the antenna gate, while the other is grounded at low temperature so as to apply RF current $I_{RF}$ with frequency $\omega_{RF}$. The input RF power was fixed at −35 dBm throughout the measurements. The transverse rf field $B_x$ is about 0.8 mT estimated from the input power and the resistance of the antenna gate. On the other hand, variable actuation voltage $V_a$ was applied to the upper Schottky gate electrode fabricated on the left clamping point using a function generator while the 2DEG beneath the gate is grounded as shown in Supplementary Fig. 1a, b. The resulting piezovoltage arising from the motion of the beam at the lower gate was amplified by a home-made HEMT-based cryogenic amplifier with the input load resistor of $R_L = 1$ kΩ followed by a commercially available amplifier (NF SA-220F5) with a total voltage gain of ~200 VV$^{-1}$. The amplified voltage signal was then measured with a lock-in amplifier phase-locked to the function generator. The details for this amplification setup are also available in refs.[39,40].

**Resistively detected NMR.** The resistively detected NMR method relies on contact hyperfine interaction between conduction electrons and nearby nuclei, thereby enabling current-induced dynamical nuclear polarization (DNP), as well as resistive detection[15,21,54–56]. In our device, a shallow mesa sustaining a 2DEG was patterned onto the right clamping point. A static magnetic field $B_0$ was then applied perpendicular to the 2DEG. Supplementary Figure 1c shows the two-terminal resistance measured as a function of $B_0$, showing the integer quantum Hall plateaus. In the quantum Hall regime, current is carried by conduction electrons flowing along two counter-propagating edge channels. In order to selectively polarize nuclear spins and detect their spin states solely at the clamping point, negative bias voltage $V_g$ is applied to the Schottky electrode to form a narrow constriction at the right clamping point as shown in Supplementary Fig. 1a, b. When a large $V_{sd}$ is applied, a large electric field owing to a high current density at the narrow constriction locally breaks down the quantum Hall state that entails spin flip-flop events between hyperfine-coupled nuclei and electrons. As a consequence of these spin flip-flop events, local nuclei at the clamping point can be spin-polarized, known as DNP[56–59]. The number of nuclei in this interaction is estimated to be ~10$^9$ or less from the effective volume of the constriction, which is roughly 1 μm × 1 μm × 20 nm.

The hyperfine field generated by the spin-polarized nuclei modifies the Zeeman energy of the electron and thereby alters the backscattering probability which is manifested as a change in $I_{sd}$. Supplementary Figure 1d shows the real time evolution of $I_{sd}$, after the abrupt application of $V_{sd} = 13.2$ mV at $B_0 = 3.5$ T ($\nu \approx 3.15$) with $V_g = -0.45$ V to induce breakdown, which reveals a slow decrease for the first 30 s followed by a saturation. The observed decrease in $I_{sd}$ can be attributed to current-induced DNP resulting in a 20% spin polarization estimated from the numerical simulations of the NMR spectra. Since this DNP was observed only when $V_g$ value is large enough to completely deplete the 2DEG underlying the electrode, it ensures that the DNP and its resistive detection are taking place solely in the vicinity of the narrow constriction, i.e., at the clamping point. After waiting

for $I_{sd}$ to saturate, $\omega_{RF}$ is slowly swept at a rate of 3 kHz min$^{-1}$. In order to resolve the fine peak structures in the NMR, $(\Delta I_{sd})^2$ is plotted as shown in Fig. 3b. When $\omega_{RF}$ matches the transition frequency of nuclear spin states, a coherent oscillation is driven that leads to the depolarization of the nuclei. The resultant change in the hyperfine field due to the depolarized nuclei modifies $I_{sd}$ that manifests itself as an NMR peak as shown in Figs. 1d and 3b. The NMR detection via the electronic transport measurements has a fluctuation of about 0.3 nA that originates from charge fluctuation in the heterostructure. Because of this current fluctuation, the signal-to-noise ratio in the NMR signal ranges from 2 to 10.

In this work, an NMR signal was observed in the magnetic field range of 3.1 < $B_0$ < 3.55 T, which corresponds to a transition from filling factor $\nu = 4$ to $\nu = 3$ quantum Hall states as highlighted in Supplementary Fig. 1c.

**Sideband NMR spectra.** The NMR spectra on the red/blue sidebands shown in Fig. 2e can in principle be obtained by calculating the time evolution via the master equation for the Hamiltonian in Eq. (1). However, the full numerical time evolution without approximation was slow to execute due to the low transition rate of the sideband NMR. Instead a numerical simulation of the Hamiltonian, approximated using the rotating-wave approximation, is employed. Equation (1) is evaluated with $x(t) = x_1 \cos(\omega_M t)$ and by moving to a non-uniform rotating frame via a unitary transformation:

$$\hat{U}_1 = \exp\left\{ i \frac{A_1 x_1}{\omega_M \hbar} \sin(\omega_M t)\left[3\hat{I}_z^2 - I(I+1)\right]/3 \right\}. \quad (3)$$

This yields a transformed Hamiltonian:

$$
\begin{aligned}
\hat{U}_1\hat{H}\hat{U}_1^\dagger - i\hbar\hat{U}_1\frac{\mathrm{d}\hat{U}_1^\dagger}{\mathrm{d}t} \simeq \ & -\hbar\omega_0\hat{I}_z - \gamma\hbar B_x\cos(\omega_{RF}t)\hat{I}_x \\
& + \frac{\gamma A_1 x_1 B_x}{\omega_M}\cos\{(\omega_0+\omega_M)t\}\hat{I}_{side} \\
& - \frac{\gamma A_1 x_1 B_x}{\omega_M}\cos\{(\omega_0-\omega_M)t\}\hat{I}_{side} \\
& + O\left(\frac{A_1 x_1}{\hbar\omega_M}\right)^2,
\end{aligned}
\quad (4)
$$

The second term in the right hand side describes the main NMR peak at $\omega_{RF} = \omega_0$, while the third (fourth) term corresponds to the blue (red) sideband. Here $\hat{I}_{side}$ describes the transition of the nuclear spin states at the sidebands and is defined as

$$\hat{I}_{side} \equiv \frac{\sqrt{3}i}{2}\left\{ \left|\tfrac{3}{2}\right\rangle\left\langle\tfrac{1}{2}\right| - \left|\tfrac{1}{2}\right\rangle\left\langle\tfrac{3}{2}\right| - \left|-\tfrac{1}{2}\right\rangle\left\langle-\tfrac{3}{2}\right| + \left|-\tfrac{3}{2}\right\rangle\left\langle-\tfrac{1}{2}\right| \right\}, \quad (5)$$

where the first (last) two terms describe the transition between $|1/2\rangle$ and $|3/2\rangle$ ($|-1/2\rangle$ and $|-3/2\rangle$) states. In order to calculate the time evolution from Eq. (4) at the red (blue) sideband, a transition to the frame rotating at frequency $\omega_{RF} \pm \omega_M$ along the z-axis is made using the transformation $\hat{U}_2 = \exp\{i(\omega_{RF} \pm \omega_M)t\hat{I}_z\}$ followed by the rotating-wave approximation to get

$$\hat{H}_{Red/Blue} \simeq -\hbar\{(\omega_0 \mp \omega_M) - \omega_{RF}\}\hat{I}_z \mp \frac{\gamma B_x A_1 x_1}{\omega_M}\hat{I}_{side}, \quad (6)$$

whose time evolution can then be easily calculated. It should be noted that, because of the rotating-wave approximation, this simulation does not reproduce the frequency shift induced by the ac strain, which is observed in Fig. 3b. Hence the main peak in Fig. 2e shows no frequency shift and this can only be reproduced in the full numerical simulation without the rotating-wave approximation as shown in Fig. 3c, d.

**Numerical simulation of the NMR spectra.** For the numerical simulations, the quadrupole coupling constants $A_1$ and $A_2$ in Eq. (2) are determined from a finite element method analysis of the strain associated with the fundamental mode (see Supplementary Note 1). We obtain $A_1 = 0.49$ meV m$^{-1}$ and $A_2 = -0.6$ meV m$^{-1}$ for $^{75}$As, $A_1 = 0.16$ meV m$^{-1}$ and $A_2 = -0.14$ meV m$^{-1}$ for $^{69}$Ga. The displacement is measured at the center of the beam (see point x in Supplementary Fig. 2a) and is decomposed into static and dynamic parts as $x(t) = x_0 + x_1 \cos(\omega_M t)$. The time evolution for the density matrix $\rho$ of the nuclear spin is then calculated using the von Neumann equation[13,57]:

$$i\hbar\frac{\mathrm{d}\rho(t)}{\mathrm{d}t} = \left[\rho(t), \hat{H}\right]. \quad (7)$$

The temporal evolution of this first-order ordinary differential equation is extracted using the fourth-order Runge–Kutta method with a time step of $\Delta t = 10^{-9}$ s. The upper panels in Supplementary Fig. 3a, b show the simulated temporal evolution of $\langle\hat{I}_z\rangle = \mathrm{Tr}\{\rho\hat{I}_z\}$ with two different mechanical displacements (a) $x_1 = 0$ and (b) $x_1 = 1.1$ μm. Both plots show the first 1.5 periods of the coherent oscillation. From this time evolution, the normalized oscillation amplitude in $\langle\hat{I}_z\rangle$ can be determined as shown in the lower panels in Supplementary Fig. 3a, b. In this simulation, the shape of the measured NMR spectra (Fig. 3b) is reproduced by adjusting $x_0$ and $\rho(0)$ values. The experimental spectrum exhibits two features: (i) the peak splitting in the absence of driven mechanical strain $x_1 = 0$ is ~5 kHz, and

(ii) the prominent peak's blue shift (solid circle) whilst the sub peak (open square) is stationary. The former feature is reproduced with $x_0 = -22$ nm, whereas the latter feature is reproduced by $\rho(0) = \{\rho_{3/2}, \rho_{1/2}, \rho_{-1/2}, \rho_{-3/2}\} = \{0.15, 0.2, 0.27, 0.36\}$, which is determined from a Boltzman distribution with a negative spin temperature of $-4$ mK. The negative temperature indicates that the nuclear spin state is population inverted due to the negative g-factor of the conduction electrons in GaAs[57]. Figure 3c displays the resultant normalized oscillation amplitude extracted from the time evolution of $\langle \hat{I}_z \rangle = \mathrm{Tr}\{\rho \hat{I}_z\}$ at different mechanical displacements.

**Floquet perturbation analysis.** The quadrupole interaction experienced by the nuclei can be described as a time-periodic perturbation that induces a level shift and thus lifts the degeneracy of the Zeeman split energy states. To quantify this level shift a Floquet perturbation analysis[60,61] of the above Hamiltonian is employed to derive the energy diagram in Fig. 4 by decomposing it to: $\hat{H} = \hat{H}_Z + x(t)\hat{J}_Q$ with $x(t) = x_0 + x_1 \cos(\omega_M t)$. In this perturbative approach the first term is the bare Hamiltonian whose eigenenergy $E_m^{(0)}$ is exactly solved as $\hat{H}_Z|m\rangle = E_m^{(0)}|m\rangle$, while the second term is treated as a periodic time-dependent perturbation that causes an energy shift for $|m\rangle$. The eigenenergies $E_m$ for the full Hamiltonian are then expanded as a perturbative series: $E_m = E_m^{(0)} + E_m^{(1)} + E_m^{(2)} + \cdots$. The striking conclusion from the Floquet theory is that the oscillating components ($\propto x_1$) have no contribution in the first order but a finite contribution in higher order. The first-order energy shift is then calculated as

$$
\begin{aligned}
E_m^{(1)} &= x_0 \langle m|\hat{J}_Q|m\rangle, \\
&= x_0 A_1 \langle m|\left[3\hat{I}_z^2 - I(I+1)\right]/3|m\rangle, \\
&= \pm x_0 A_1 \\
&\equiv \pm \delta,
\end{aligned}
\tag{8}
$$

with a positive (negative) sign for $|\pm 3/2\rangle$ ($|\pm 1/2\rangle$). Here only the diagonal part (real part) of $\hat{J}_Q$ contributes to the energy shift. Similarly, the second-order energy shift is calculated as

$$
\begin{aligned}
E_m^{(2)} = x_0^2 &\sum_{m' \neq m} \frac{\langle m|\hat{J}_Q|m'\rangle\langle m'|\hat{J}_Q|m\rangle}{E_m - E_{m'}} \\
+ \frac{x_1^2}{4} &\sum_{m' \neq m} \frac{\langle m|\hat{J}_Q|m'\rangle\langle m'|\hat{J}_Q|m\rangle}{E_m - E_{m'} - \hbar\omega_M} \\
+ \frac{x_1^2}{4} &\sum_{m' \neq m} \frac{\langle m|\hat{J}_Q|m'\rangle\langle m'|\hat{J}_Q|m\rangle}{E_m - E_{m'} + \hbar\omega_M}.
\end{aligned}
\tag{9}
$$

The numerators in this equation can be evaluated as

$$
\begin{aligned}
\langle m|\hat{J}_Q|m'\rangle\langle m'|\hat{J}_Q|m\rangle &= -A_2^2 \langle m|\left[\hat{I}_+^2 - \hat{I}_-^2\right]|m'\rangle\langle m'|\left[\hat{I}_+^2 - \hat{I}_-^2\right]|m\rangle \\
&= 12 A_2^2 \delta_{m', m\pm 2},
\end{aligned}
\tag{10}
$$

and are independent of $m$. In the second order, the off-diagonal part (imaginary part) of $\hat{J}_Q$ has a non-zero contribution. This leads to

$$
\begin{aligned}
E_m^{(2)} &= \mp \frac{6x_0^2 A_2^2}{\hbar\omega_0} \\
&\mp \left[\frac{3x_1^2 A_2^2}{2\hbar\omega_0 + \hbar\omega_M} + \frac{3x_1^2 A_2^2}{2\hbar\omega_0 - \hbar\omega_M}\right] \\
&\approx \mp \frac{3x_1^2 A_2^2}{\hbar\omega_0} \\
&\equiv \mp \chi,
\end{aligned}
\tag{11}
$$

where the negative (positive) sign is for $|3/2\rangle$, and $|1/2\rangle$ ($|-3/2\rangle$, and $|-1/2\rangle$) with approximately: $x_0/x_1 \approx 0$, and $\omega_M/\omega_0 \approx 0$.

**Calibration of mechanical displacement and strain.** In the experiments, the mechanical motion can be detected via the piezovoltage $V_d$, from which the corresponding displacement $x_1$ at the center of the beam resonator (point $x$ in Supplementary Fig. 2a) is determined via $x_1 = V_d/(5 \times 10^3 \text{ V m}^{-1})$. This calibration was derived by fitting the experimentally observed NMR frequency shift to the simulated frequency shift as shown in Fig. 3d. The electromechanical strain can be determined from this displacement calibration via a finite element method simulation (see Supplementary Note 1). To ensure validity of this calibration, the mechanical displacement is also quantified from the onset of nonlinearity. The onset of nonlinearity in a doubly clamped beam resonator can be estimated analytically from $x_c \approx t\sqrt{2/0.528Q(1-\nu^2)} \approx 12$ nm, where $t = 1$ μm is the thickness of the beam and $\nu = 0.31$ is the Poisson ratio of GaAs[41,52]. Experimentally, nonlinearity emerges at $V_d \sim 0.2$ mV as shown in Fig. 1b and in the strong driving regime at point 5 in Fig. 3a, the mechanical amplitude is roughly 30 times larger than this onset, and is estimated to be $30x_c \approx 0.4$ μm.

**Data availability.** The data that support the findings of this study are available from the corresponding author on request.

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

## Acknowledgements

The authors thank H. Chudo, K. Chida, K. Harii, Y. Hirayama, J. Ieda, K. Koshino, Y. Matsuzaki, W. J. Munro, and M. Ono for fruitful discussions. This work was partly supported by JSPS KAKENHI Grant No. 23241046, JP18H01156, and JP18H05258 and Grant-in-Aid for Scientific Research on Innovative Areas No. JP15H05869.

## Author contributions

Y.O. fabricated the device, performed the measurements, and analyzed the data and theoretical model. K.O. grew the GaAs heterostructure. S.S. supported the device fabrication. S.N. and N-H.K supported the numerical simulation. Y.O. I.M., and H.Y. wrote the paper.

## Additional information

**Competing interests:** The authors declare no competing interests.

