## [Peer Review File · Nature Communications]

Reviewers' comments:

Reviewer #1 (Remarks to the Author):

Yuma Okazaki and coworkers report on experimental studies, where an advanced nanoscale semiconductor device was fabricated to include a high-quality mechanical resonator, a radio frequency antenna to manipulated nuclear spins, and a 2DEG device for hyperpolarization and resistive detection of nuclear magnetic resonance effects. The experimental findings are technically sound and presented in a clear manner. The results and interpretation are valid, and I have only few minor comments on the technical side of this work.

The findings of this work can be divided into two parts. The first (longer) part concerns NMR spectroscopy in presence of strong off-resonant mechanical oscillations. Mechanical driving is found to produce quadrupolar shifts, which the authors reproduce with a theoretical model. While these results (and interpretation) are convincing, I do not feel they are particularly unexpected. Each nuclear spin forms a four-level system, whose spectrum is described by the well known Zeeman and quadrupolar Hamiltonians, that can be solve using standard perturbation methods or numerics. There is not much new in terms of physics here - this is a standard problem in NMR and NAR (nuclear acoustic resonance) spectroscopy successfully applied to a variety of crystalline and powder structures with different electric field gradient profiles.

The second (much shorter) part reports on observation of side-band NMR peaks. I think this is a very interesting and novel result, as it demonstrates that nuclear-phonon system can be engineered to operate in a resolved-side-band regime. This opens prospects (at least potentially) for achieving non-trivial effects such as cooling of the resonator mechanical motion. Unfortunately, only one paragraph is dedicated to this result and then the paper ends abruptly.

My overall opinion is that a considerable effort would be needed to make this work suitable for Nature Communication in terms of novelty of the results. Ideally, more experiments exploiting the novel regime of a system with well-resolved phonon-sideband NMR spectrum would be needed. Or at least a proper discussion of the novel part of the findings and their implications is necessary. And I would recommend reducing the discussion in the first less novel part (discussion of quadrupolar NMR shifts under oscillating strain). Alternatively, the paper is suitable for a more technical journal with minor improvements.

I now list some detailed comments:

1. The red sideband of Ga71 appears to have smaller linewidth than the blue one – is it a real effect, or limited signal to noise ratio? If effect is real, what is the origin?
2. Did the authors attempt mechanical motion cooling via excitation of the red sideband? If such cooling is possible, is there a realistic method for monitoring the resulting temperature of the mechanical oscillations? If currently not possible, what are the limitations and realistic solutions?
3. End of section D in Methods. The form of the density matrix $\rho(0)$ appears to be unphysical. One of the diagonal components is 0, which strictly speaking could be achieved only for 100% polarization. How was this $\rho(0)$ derived? One would expect a Boltzmann distribution. I suggest using a full Boltzmann $\rho(0)$. Alternatively, if the component proportional to unity matrix is removed from ρ , it should be removed completely and the resulting deviation density matrix should be traceless.
4. Section E in Methods. “To ensure validity of this calibration, we quantify the mechanical displacement from the onset of nonlinearity at the critical displacement”. The calibration procedure is not entirely clear. Was nonlinearity reproduced in FEM simulations (and parameters adjusted to match the experimentally observed nonlinearity)? Please clarify.
5. The device has a “fundamental resonance frequency of 1.7 MHz”. Was this frequency chosen for a reason? What are the limitations in achieving higher frequency? Presumably, mechanical frequencies comparable to NMR or NQR frequencies would be of more interest? Are there any fundamental obstacles in designing a higher frequency device.

Several potential implications are mentioned in the abstract and introduction, but lack proper justification and look at the moment as overclaims.

6. “entanglement between sound and nuclei”. This might be possible in principle, but can the authors discuss how feasible it is? I immediately see some obstacles in achieving this. At least with the current device design. For example: the nano-beam consists of a very large number of nuclei, but only a very small fraction of nuclei is controlled and measured using NMR. In other words the “coupling strength” between the phonon modes and the “modes” of the probed nuclei is small, while the phonon modes are strongly coupled to a much larger number of uncontrollable nuclei. How do the authors envisage overcoming this problem? How large is the fraction of the spins that needs to be polarized driven with rf for entanglement to work in principle.
7. Besides, nuclear spin bath is a many-body system. So this system is fundamentally different from atom-in-a-cavity system. How entanglement is meant to work in this scenario?

8. “nuclear spin freezing” – I don’t really see how this is possible. Presumably, this would require achieving a mechanical oscillation spectrum with well resolved nuclear-spin-flip assisted sideband. Any realistic proposal to achieve this? What are the criteria to achieve such a regime?

9. “so that non-adiabatic dynamical effects such as the ac-Stark shift and side-band transitions of nuclear spin states can be accessed.” The ac-Stark part is not demonstrated here.

More comments.

10. In the introduction “Conversely this also implies great difficulty in manipulating their state via external parameters.” I am not sure what is implied here. I don’t see what the difficulty is – using radio-frequency fields one can perform arbitrary rotations of the nuclear spin ensemble wavefunction on demand. Please remove or clarify.

11. “Specifically the resultant NMR exhibits frequency shifts in response to the dynamical strain which even exceeds their Zeeman energy splitting.” Sentence not clear. The strain induced quadrupolar shifts are tens of kHz, while Zeeman splitting (NMR frequency) is tens of MHz. Please rectify.

Reviewer #2 (Remarks to the Author):

The paper of Okazaki et al., describes strain-mediated manipulation of nuclear spins in a two-dimensional gas in GaAs. They resistively detect nuclear magnetic resonance signals in a quantum hall device and apply dynamical strain by a micromechanical resonator clamped to the nuclear spin ensemble. They observe AC shifts induced on the nuclei by the oscillating strain. They finally demonstrate that the spins can be driven into the resolved sideband regime.

The result is remarkable and will certainly be of interest to a large readership in the communities of both quantum hall physics and quantum devices. As such I recommend it for publication, provided that the authors can address the following remarks:

Motion in RF gradient: the nuclei are sitting very close to the wire supplying the RF drive for magnetic resonance, so they are likely subjected to a considerable magnetic field gradient. Could motion in this gradient, induced by the oscillation of the resonator, mimick the effects that the authors assign to their quadrupole moment interacting with strain?

Sample volume: the authors state that NMR can only be detected in the constriction created by the gate wire. Most of this constriction is situated far from the mechanical resonator in a region

of the substrate where I would not expect any strain. Why is there no larger background induced by nuclei in this region and/or a smearing of the mechanical sidebands?

Scope: driving into the resolved sideband regime in my opinion is a key finding of the study. I would suggest to highlight it more at an earlier stage by expanding the claim in the abstract or at least the introduction. Also, I believe that some more comments on the state of the art would be helpful. In which other systems has the resolved-sideband-regime been reached so far? What kind of applications can be envisaged? This could deserve as much or even more attention than the aspect of how to control nuclear spins, which in my opinion is not as difficult as the authors make it appear.

The same applies to the implications of the study. The outlook of the present manuscript is rather short and restricted to possible improvements of the device. I would be interested whether mechanical cooling/freezing of the nuclear spins as suggested by the authors is a realistic prospect. What would be the cooling rate in such a device? Which other materials could display similar effects? Is there for instance a chance to use NMR for ultrasound detection in soft matter?

I also have a couple of more detailed comments:

Abstract: the authors could expand a bit on the term "nuclear spin freezing"

p.2 Why is the Rabi frequency the relevant scale that off-resonant driving needs to overcome in order to create shifts? Shouldn't that rather be some $T_1/T_2/T_2^*/\dots$ scale

p.2 "exceeds" should be "exceed"

p. 3 "from GaAs/AlGaAs heterowafers" - there is an "a" missing

p. 3/Methods C, being an outsider I do not understand how the authors measure the NMR spectra. Why are the electrons spin-polarized when the quantum Hall effect breaks down. Is the operation at the edge of a quantum Hall step essential for NMR detection? It also would be good to estimate the degree of spin polarization. There is an "is" missing in C. p. 8

p.3 "under the influence of intense strain". It is not clear from the text whether the strain is DC or AC.

p.3 "at zero mechanical displacement with a doublet peak structure" - why should it be a doublet structure at zero strain?

p. 4 "asymmetrically split" - the splitting looks symmetrical in Fig. 3

p.5 "novel effects" - how novel are they? Is it the first observation of resolved sidebands in NMR?

Methods B. - the magnetic field should be given as well as the part numbers of the amplifiers.

Methods D. does the meter in " $A_1 = .49 \text{ meV/m}$ " refer to displacement at the center or at the edge of the resonator?

Fig. 1 - antenna is mis-spelled in the caption and the figure. The order of magnitude of the magnetic field should be given. It is not clear where the current sourced by voltage source V_a is flowing, since there is no ground closing the circuit. It is not clear where voltage V_d is referenced, since there is only one line going into the amplifier. Fig c) suggests that the authors are sourcing a current across the constriction while the methods section states that they applied a voltage and merely measured the current. It is not clear how the squared quantity ΔI_{sd}^2 can drop below zero in 1d.

Fig. 2. Why does the oscillation amplitude rise with detuning rather than dropping from a maximum at zero detuning? Is there a reason why the signal ratio differs between the experiment and the theory?

Reviewer #3 (Remarks to the Author):

This is an impressive experiment that I am not at all surprised was carried out at NTT. I commend the authors for having undertaken that experiment, and brought the electrically detected NMR to that level, i.e. to combine the NMR with dynamical strain. I have seriously thought in the past about making this experiment myself, so when I was asked to be the reviewer I was at first skeptical regarding the data and the results claimed. Proper reading of the manuscript by my student and I proved the work to be clear, systematic, well-engineered, and its interpretation well thought out. A non-specialist in the field could criticize the line shapes shown and could find them departing from what we expect from "traditional" NMR. But here, the signal-to-noise is sufficient, and the effect of the dynamical strain appears clear to my eye. The simulation also rings to me as correct. The data, methods and analysis are all complete, coherent and most importantly new. The technique developed and described in the paper is similar in reasoning to other ones that require nuclear spins of long relaxation time, however it goes further by utilizing the dynamic strain induced manipulation of the quadrupole interaction. Finally, this is a somewhat across the field paper that fits extremely well within the nature comm format and its mission. It bridges different aspect of physics, and also different communities. It is clear to me that it can be published in Nature Comm. The only caveat is that I am not yet convinced that

this paper will become highly cited rapidly and this is because it is somewhat a “heroic effort” that will take time for others to catch up. This being said, I strongly feel this should be published in Nature Comm with any further delay.

I have some points below (some more important than others) that I wish the authors will consider before going to print. They are not obliged to agree with them, but as a friendly competitor in the field, I hope to receive a response from the authors where due diligence has been done.

- On page 3, the authors discuss the motional piezo voltage measured for different actuation voltages, and refer to figures 1b and 2a, although 2a does not seem relevant for this information.
- On page 10, when developing the perturbation theory terms, the authors never mention why they neglect the real or imaginary parts in certain calculations; in fact Eq. 4 completely omits the imaginary part of the quadrupole operator whereas Eq. 6 omits the real part, without any explanation or justification. When looking at the diagonal and off-diagonal terms in the matrix of the Supp. Info, it becomes a bit clearer, however this should be made explicit to the readers.
- There is no discussion or characterization of errors of any kind. This seems very unfortunate; when developing a new technique such limitations should be addressed to a certain degree, especially if they are to be considered negligible at all (which I suppose is the case seeing as how they are not even visible in the plots).
- In some sentences, the use of parentheses for "(typically > MHz)" seems unnecessary, though I find the way this part was written a little off-putting.
- On numerous occasions the authors refer the reader to the "Method" section by writing "see Method". This should be written as "Methods" or "Method section".
- Figure 2d shows two separate curves with different color data points, referred to as "simulation" and "experiment" by little lines of corresponding color; this seems inefficient and would be better with a standard legend.
- The second paragraph on page 5 starts with a run-on sentence; a comma needs to be put somewhere or it needs to be split into two different sentences to be clear.
- The curves in blue and orange of figures 4c-d with "x3" indicating the amplitude was multiplied by three (to be more visible to the reader) is odd; a second y-axis would suffice (and still no uncertainty values are visible!).
- The use of "On the other hand" towards the end of the second paragraph on page 5 is out of place and does not follow with a subject contradicting the logic of the phrase preceding it.
- The phrasing "For more advanced dynamic quantum manipulation of nuclear spins via electromechanical phonons requires stronger coupling between them" needs some work and refining.
- In Figure 1, "Mechanical motion" is not capitalized at all, let alone the same way as the other titles in the figures. The same goes for a couple of subtitles in figure 3.

Typos:

- Page 7, paragraph 2: "these spin flip-flop events".
- Page 7, paragraph 3: "Figs. 1d and 2b." (No comma after 2b)
- Page 9, first paragraph: "Figs. S3a and S3b" (note: this occurs twice in the same paragraph).
- Page 9, first paragraph: " $x_1 = 0$ " (no comma after the "0").
- Page 9, paragraph 2: "the oscillating components (...) have no contribution in the first

Responses to Reviewers' Comments:

Reviewer #1

Q1: "Yuma Okazaki and coworkers report on experimental studies, where an advanced nanoscale semiconductor device was fabricated to include a high-quality mechanical resonator, a radio frequency antenna to manipulated nuclear spins, and a 2DEG device for hyperpolarization and resistive detection of nuclear magnetic resonance effects. The experimental findings are technically sound and presented in a clear manner. The results and interpretation are valid, and I have only few minor comments on the technical side of this work.

The findings of this work can be divided into two parts. The first (longer) part concerns NMR spectroscopy in presence of strong off-resonant mechanical oscillations. Mechanical driving is found to produce quadrupolar shifts, which the authors reproduce with a theoretical model. While these results (and interpretation) are convincing, I do not feel they are particularly unexpected. Each nuclear spin forms a four-level system, whose spectrum is described by the well known Zeeman and quadrupolar Hamiltonians, that can be solve using standard perturbation methods or numerics. There is not much new in terms of physics here - this is a standard problem in NMR and NAR (nuclear acoustic resonance) spectroscopy successfully applied to a variety of crystalline and powder structures with different electric field gradient profiles.

The second (much shorter) part reports on observation of side-band NMR peaks. I think this is a very interesting and novel result, as it demonstrates that nuclear-phonon system can be engineered to operate in a resolved-side-band regime. This opens prospects (at least potentially) for achieving non-trivial effects such as cooling of the resonator mechanical motion. Unfortunately, only one paragraph is dedicated to this result and then the paper ends abruptly.

My overall opinion is that a considerable effort would be needed to make this work suitable for Nature Communication in terms of novelty of the results. Ideally, more experiments exploiting the novel regime of a system with well-resolved phonon-sideband NMR spectrum would be needed. Or at least a proper discussion of the novel part of the findings and their implications is necessary. And I would recommend reducing the discussion in the first less novel part (discussion of quadrupolar NMR shifts under oscillating strain). Alternatively, the paper is suitable for a more technical journal with minor improvements."

A1: The authors thank the reviewer for the careful reading of the manuscript.

Following Reviewer #1's comment "a proper discussion of the novel part of the findings and their implications is necessary", which recommends us to highlight the observation of the sideband NMR more, the title, abstract and the introduction have been completely revised to highlight the importance of mechanical sidebands in NMR spectra as well as the first observation of mechanically assisted sideband transitions in NMR. Further a theoretical model and numerical simulations have also been introduced to explain the physics of the mechanically aided sideband transitions NMR. Moreover an estimation of the conditions needed for the strong coupling regime is added to the discussion part.

Reviewer #1 was of the opinion that NMR frequency shift induced by a higher-order quadrupole term is a standard problem in NMR and NAR and thus recommends us to reduce the discussion on the observation of the frequency shifts. In these experiments, however, frequency shifts induced by rapidly oscillating strain are extremely challenging with bulk specimens as it is difficult to achieve the requisite strain ($\sim 10^{-3}$) needed for this observation. In contrast here with the aid of the mechanical resonator, which can enhance strain by the product of the quality factor ($Q > 10^4$), enables access to an unprecedented strain regime to observe this effect. Additionally, in a recent report of a mechanical resonator and superconducting qubit hybrid system [Pirkkalainen et al. Nature 494, 211 (2013)], both qubit frequency shifts induced by the driven mechanical oscillation as well as sideband transitions were observed and explained in terms of dynamical coupling between the mechanics and qubit. The present work is analogous to this report where the 2-level qubit manifold is now replaced with the Zeeman split nuclei manifold. In the revised manuscript parallels are drawn with this work to enable the reader to contextualize the present work's novelty and the importance of the NMR frequency shifts induced by mechanical oscillations.

Q1-1: The red sideband of Ga71 appears to have smaller linewidth than the blue one? Is it a real effect, or limited signal to noise ratio? If effect is real, what is the origin?

A1-1: This is due to the limited signal to noise ratio and is addressed in the revised manuscript.

Q1-2: Did the authors attempt mechanical motion cooling via excitation of the red sideband? If such cooling is possible, is there a realistic method for monitoring the resulting temperature of the mechanical oscillations? If currently not possible, what are the limitations

and realistic solutions?

A1-2: The temperature of the mechanical oscillation can be determined by measuring the corresponding thermal motion (Brownian motion) via a quantum point contact or quantum dot being used as a displacement sensor where this was demonstrated in earlier work [Okazaki et al. Appl. Phys. Lett. 103, 192105 (2013), and Nature Communications 7, 11132 (2016)].

However in the original manuscript the author's intention was to cool the nuclear spins via the mechanical resonator rather than the other way around. Such electromechanical or acoustic cooling of nuclear spins requires a high-frequency mechanical resonator as well as strong coupling between spins and mechanics. In the revised manuscript the authors have clarified these points in both the abstract and the discussion section.

Q1-3: End of section D in Methods. The form of the density matrix $\rho(0)$ appears to be unphysical. One of the diagonal components is 0, which strictly speaking could be achieved only for 100% polarization. How was this $\rho(0)$ derived? One would expect a Boltzmann distribution. I suggest using a full Boltzmann $\rho(0)$. Alternatively, if the component proportional to unity matrix is removed from ρ , it should be removed completely and the resulting deviation density matrix should be traceless.

A1-3: $\rho(0)$ in the original manuscript was artificially set to reproduce the observed NMR spectra. In previous reports on resistively detected NMR [Yusa et al. Nature 434, 1001 (2005), T. Ota et al. APL 90, 102118 (2007)], a Boltzmann distribution with a negative spin temperature was assumed as the initial condition for $\rho(0)$, where the nuclear spins can be population inverted because of the negative g-factor of conduction electrons in GaAs. As the reviewer suggested, in the revised manuscript, a Boltzmann distribution is employed with $\rho(0) = \{0.15, 0.2, 0.27, 0.36\}$ and the corresponding spectra are recalculated. The revised $\rho(0)$ corresponds to a negative spin temperature of -4 mK. Although the resultant shapes of the spectra are somewhat modified, the central observation of frequency shifts being induced by the mechanical motion still remains.

Q1-4: Section E in Methods. "To ensure validity of this calibration, we quantify the mechanical displacement from the onset of nonlinearity at the critical displacement". The calibration procedure is not entirely clear. Was nonlinearity reproduced in FEM simulations (and parameters adjusted to match the experimentally observed nonlinearity)? Please

clarify.

A1-4: According to [Ekinici et al. Rev. Sci. Instrum. 76, 061101 (2005)], the onset of nonlinearity can be estimated analytically from the thickness of the beam resonator, its quality factor and Poisson's ratio. This analytical expression has previously been used to estimate displacement order of a beam resonator [Mahboob et al. Nature Nano 3, 275 (2007)].

The nonlinearity of our resonator is small as the nonlinear correction to the mechanical resonant frequency is about 13 kHz, which is smaller than the bare resonance frequency of 1.7 MHz. Hence our linear FEM simulation, which cannot reproduce this nonlinearity, is sufficiently accurate to enable faithful determination of the resultant strain tensor.

Q1-5: The device has a “fundamental resonance frequency of 1.7 MHz”. Was this frequency chosen for a reason? What are the limitations in achieving higher frequency? Presumably, mechanical frequencies comparable to NMR or NQR frequencies would be of more interest? Are there any fundamental obstacles in designing a higher frequency device?

A1-5: The 1.7 MHz frequency was selected for technical reasons due to the limited bandwidth of the home-made cryo-amplifier. However in this device higher-order modes up to a few tens of MHz could also be observed. Practically it is possible to access a higher frequency regime with a doubly clamped mechanical resonator with a fundamental frequency of several tens of MHz [Pirkkalainen et al. Nature 494, 211 (2013)] or with a bulk acoustic mechanical resonator [Connel et al. Nature 464, 697 (2010)] to access the GHz regime. When the mechanical resonator is on-resonant with NMR/NQR, phonon dressed states of the nuclear spins would be formed. In this case more advanced quantum control such as a state transfer between the mechanical resonator and nuclear spin ensemble i.e. Rabi oscillations would be possible. In the revised manuscript such future possibilities are noted with higher frequency mechanical resonators and the means to realistically implement them.

Q1-6: Several potential implications are mentioned in the abstract and introduction, but lack proper justification and look at the moment as overclaims.

A1-6: Accessing the strong-coupling regime is the key to realizing advanced experiments such as the sideband cooling and the coherent coupling mentioned in the introduction. In

order to justify these possibilities in the revised manuscript, the means to increasing the coupling strength between mechanics and nuclei are addressed. Based on this estimation for the coupling suggests that the strong coupling regime should be attainable in the future with this platform thus making potential implications of this work more realistic.

Q1-7: “entanglement between sound and nuclei”. This might be possible in principle, but can the authors discuss how feasible it is? I immediately see some obstacles in achieving this. At least with the current device design. For example: the nano-beam consists of a very large number of nuclei, but only a very small fraction of nuclei is controlled and measured using NMR. In other words the “coupling strength” between the phonon modes and the “modes” of the probed nuclei is small, while the phonon modes are strongly coupled to a much larger number of uncontrollable nuclei. How do the authors envisage overcoming this problem? How large is the fraction of the spins that needs to be polarized driven with rf for entanglement to work in principle.

A1-7: As the reviewer pointed out, nuclear spins are homogeneously distributed into the mechanical resonator to which they can couple with a coupling strength proportional to the distribution of strain. This unintentional coupling to unpolarized nuclear spins can be mitigated by using the electrically detected NMR technique where only local nuclear spins are selectively polarized, brought into resonance and then detected. Consequently only the correlations between this local ensemble of nuclear spins and the electromechanical resonator are sampled. Crucially this experiment requires strong coupling which could be achieved when the resonator interacts with 10^{10} polarized nuclear spins as estimated in the discussion section in the revised manuscript. In order to polarize and detect this number of nuclear spins would require a redesign of the transport channel and engineering of the motional strain to coincide with it where both of these requirements are achievable with present technology.

Q1-8: Besides, nuclear spin bath is a many-body system. So this system is fundamentally different from atom-in-a-cavity system. How entanglement is meant to work in this scenario?

A1-8: As the reviewer pointed out, the coupling between a single nuclear spin and a mechanical resonator is quite challenging. Alternatively, the coupling to an ensemble of nuclear spins is possible in which a collective motion of the nuclear spins is entangled with the resonator. Indeed such coupling has been demonstrated with an electron spin ensemble [Kubo et al. Phys. Rev. Lett. 107, 220501 (2011) or Zhu et al. Nature 478, 221 (2011)] and

these references have been added to the revised manuscript.

Q1-9: “nuclear spin freezing” ? I don’t really see how this is possible. Presumably, this would require achieving a mechanical oscillation spectrum with well resolved nuclear-spin-flip assisted sideband. Any realistic proposal to achieve this? What are the criteria to achieve such a regime?

A1-9: “nuclear spin freezing” has been omitted in the revised manuscript. To cool the nuclear spin ensemble via the mechanics the resolved sideband regime would be needed as well as strong coupling between the sub-systems. In the discussion section in the revised manuscript these points are addressed in particular the ensemble enhancement of the effective coupling strength: $\sqrt{N}g_0$ when $N \sim 10^{10}$ nuclear spins. This requisite number of nuclear spins can be accessed in realistic future experiments.

Q1-10: “so that non-adiabatic dynamical effects such as the ac-Stark shift and side-band transitions of nuclear spin states can be accessed.” The ac-Stark part is not demonstrated here.

Q1-10” The observation of NMR frequency shift induced by the driven mechanical oscillation [Figs. 3 and 4] demonstrates the mechanical analogue of the ac-Stark shift with the nuclei. The ordinary ac-Stark shift is a frequency shift in atomic transitions induced by the rapidly oscillating electric field. In a case where the frequency of the electric field is detuned from the atomic transition, the oscillating field can be treated as higher-order perturbation which induces a quadratic frequency shift which is referred to the ac-Stark shift in the dispersive regime and is routinely observed in circuit QED architectures [“Dispersive regime of circuit QED”, Boissonneault et al. PRA 79, 013819 (2009); “ac Stark Shift and Dephasing of a Superconducting Qubit Strongly Coupled to a Cavity Field”, D. I. Schuster et al., Phys. Rev. Lett. 94, 123602 (2005)]. In the present experiment, the role of the oscillating electric field is played by the oscillating mechanical strain. Indeed this mechanical analogue of the ac-Stark effect has also been observed in an electromechanical resonator-superconducting qubit hybrid system [Pirkkalainen et al. Nature 494, 211 (2013)] where the qubit frequency shifted due to the driven electromechanical resonator whose frequency was far detuned from the qubit’s transition frequency. Indeed Pirkkalainen et al. also analyze the frequency shift, which they refer to as the mechanical/phonon ac-Stark shift, within the Floquet framework. In the revised manuscript these papers are cited and this analogy is better developed.

Q1-11: In the introduction “Conversely this also implies great difficulty in manipulating their state via external parameters.” I am not sure what is implied here. I don’t see what the difficulty is ? using radio-frequency fields one can perform arbitrary rotations of the nuclear spin ensemble wavefunction on demand. Please remove or clarify.

A1-11: The introduction has been revised and the above sentence has been omitted.

Q1-12: “Specifically the resultant NMR exhibits frequency shifts in response to the dynamical strain which even exceeds their Zeeman energy splitting.” Sentence not clear. The strain induced quadrupolar shifts are tens of kHz, while Zeeman splitting (NMR frequency) is tens of MHz. Please rectify.

A1-12: This sentence has been omitted in the revision.

Reviewer #2

Q2: "The paper of Okazaki et al., describes strain-mediated manipulation of nuclear spins in a two-dimensional gas in GaAs. They resistively detect nuclear magnetic resonance signals in a quantum hall device and apply dynamical strain by a micromechanical resonator clamped to the nuclear spin ensemble. They observe AC shifts induced on the nuclei by the oscillating strain. They finally demonstrate that the spins can be driven into the resolved sideband regime.

The result is remarkable and will certainly be of interest to a large readership in the communities of both quantum hall physics and quantum devices. As such I recommend it for publication, provided that the authors can address the following remarks:"

A2: The authors thank the reviewer for the careful reading of the manuscript. We revised the manuscript based on the Reviewer #2's comments as detailed below.

Q2-1: Motion in RF gradient: the nuclei are sitting very close to the wire supplying the RF drive for magnetic resonance, so they are likely subjected to a considerable magnetic field gradient. Could motion in this gradient, induced by the oscillation of the resonator, mimic the effects that the authors assign to their quadrupole moment interacting with strain?

A2-1: This effect is at least 4 orders of magnitude smaller than the nuclear interaction induced by the motional strain and can be estimated as follows. The change in the distance between the antenna gate and the nuclei induced by the mechanical strain is of the order of 10^{-3} or about 0.1 nm. The corresponding change of the transverse magnetic field due to gradient would then be of the order of micro tesla which gives the resultant Zeeman term a contribution of about 10 Hz, which is negligible.

Q2-2: Sample volume: the authors state that NMR can only be detected in the constriction created by the gate wire. Most of this constriction is situated far from the mechanical resonator in a region of the substrate where I would not expect any strain. Why is there no larger background induced by nuclei in this region and/or a smearing of the mechanical sidebands?

A2-2: In the present work the NMR (polarization and) detection is based on the contact interaction between nuclear spins and the conduction electrons in the two-dimensional

electron gas (2DEG). Consequently only the nuclei that interact with the electrons can be detected. Since the thickness of the 2DEG is about 10–20 nm and is located 100 nm below the surface of the device, only the nuclei in this local region are accessed in this system and the nuclei in the substrate have no contribution in our measurement. To address this point in the revised manuscript, a schematic of the side view of the device is introduced in Fig. S1b.

Q2-3: Scope: driving into the resolved sideband regime in my opinion is a key finding of the study. I would suggest to highlight it more at an earlier stage by expanding the claim in the abstract or at least the introduction. Also, I believe that some more comments on the state of the art would be helpful. In which other systems has the resolved-sideband-regime been reached so far? What kind of applications can be envisaged? This could deserve as much or even more attention than the aspect of how to control nuclear spins, which in my opinion is not as difficult as the authors make it appear.

A2-3: The introduction has been revised to address this point with the side-band physics described to contextualize the observations of NMR with mechanically assisted side-bands.

Q2-4: The same applies to the implications of the study. The outlook of the present manuscript is rather short and restricted to possible improvements of the device. I would be interested whether mechanical cooling/freezing of the nuclear spins as suggested by the authors is a realistic prospect. What would be the cooling rate in such a device?

Which other materials could display similar effects? Is there for instance a chance to use NMR for ultrasound detection in soft matter?

A2-4: In order to achieve cooling the strong coupling regime would be needed. In this limit from [Grajcar PRB 78, 035406 (2008)] a lower limit on the achievable temperature in resonator-based sideband cooling is given by $T = \hbar\omega_{NS}/\hbar\omega_R T_R$, where $\hbar\omega_{NS}$ is the transition frequency of nuclear spins, $\hbar\omega_R$ is the resonator frequency, and T_R is the ambient temperature. Assuming $\hbar\omega_{NS} = 2\pi \times 25$ MHz, $\hbar\omega_R = 1$ GHz and $T_R = 10$ mK potentially being available in a future experiment, cooling below 1 mK would be possible.

Nuclear spins with $I > 1/2$ are known as the quadrupole nuclei and have a non-zero quadrupole interaction. Other material containing quadrupole nuclei in principle can display similar effects. The authors speculatively anticipate ultrasound detection being possible for single crystals but detection in soft matter may be difficult.

Q2-5: Abstract: the authors could expand a bit on the term "nuclear spin freezing"

A2-5: The authors have eliminated "nuclear spin freezing" in the revised manuscript.

Q2-6: p.2 Why is the Rabi frequency the relevant scale that off-resonant driving needs to overcome in order to create shifts? Shouldn't that rather be some $T_1/T_2/T_2^*/\dots$ scale

A2-6: Naturally the mechanical frequency needs to be faster than $1/T_1$, $1/T_2$ etc. However it is also important to be faster than the Rabi frequency of the NMR. If the mechanical frequency was slower than the Rabi frequency of the NMR transition, dynamical effects such as the frequency shift [Figs. 3 and 4] would not be observed. Instead a broadening of the NMR spectrum due to the time-averaging of the slowly varying quadrupole shift associated with the mechanical motion would be observed.

Q2-7: p. 3/Methods C, being an outsider I do not understand how the authors measure the NMR spectra. Why are the electrons spin-polarized when the quantum Hall effect breaks down. Is the operation at the edge of a quantum Hall step essential for NMR detection? It also would be good to estimate the degree of spin polarization. There is an "is" missing in C. p. 8

A2-7: In the integer quantum Hall with an odd filling factor, the conduction electrons in the edge channel are 100% spin polarized due to Zeeman splitting. For example, at $B = 3.2$ T, the Zeeman splitting of the conduction electrons is about 160 μeV (~ 2 K), which is larger than the system temperature ~ 80 mK. In order to polarize nuclear spins, a transfer of angular momentum from spin-polarized electrons to the nuclei is necessary. When a large current is introduced into a narrow constriction, the quantum Hall effect is locally broken down which entails a tunneling between the spin-up and spin-down Landau levels. Because of this spin-flip tunneling events, the spin polarized electrons can transfer their angular momentum to the nuclei, and thus the local nuclei at the constriction can be selectively polarized. The mechanism of the resistive detection is similar, i.e. the change in the polarization changes the resistance. The spin degree of polarization is estimated to be 20 % from the numerical simulations of the NMR spectra.

Q2-8: p.3 "under the influence of intense strain". It is not clear from the text whether the strain is DC or AC.

A2-8: This sentence was modified to “under the influence of intense AC strain”.

Q2-9: p.3 "at zero mechanical displacement with a doublet peak structure" - why should it be a doublet structure at zero strain?

A2-9: Even in the absence of the mechanical motion (zero AC strain), the nuclear spins are subjected to unintentional DC strain due to residual strain in the heterostructure. This point further addressed in the revised manuscript on page 6~7 and in the caption to Fig. 4.

Q2-10: p. 4 "asymmetrically split" - the splitting looks symmetrical in Fig. 3

A2-10: The term “asymmetrically” is now omitted.

Q2-11: p.5 "novel effects" - how novel are they? Is it the first observation of resolved sidebands in NMR?

A2-11: To the author’s best knowledge this is the first observation of the resolved sideband NMR induced by mechanical oscillations. In the revised manuscript this point is highlighted in the introduction.

Q2-12: Methods B. - the magnetic field should be given as well as the part numbers of the amplifiers.

A2-12: The magnetic field value $B_x = 0.8$ mT and the part number for the amplifier NF SA-220F5 are now described in the revision.

Q2-13: Methods D. does the meter in " $A_1 = .49$ meV/m" refer to displacement at the center or at the edge of the resonator?

A2-13: The displacement is measured at the center of the beam as indicated by ‘x’ in Fig. S2a. This point included in the revised manuscript as “the displacement is measured at the center of the beam resonator.”

Q2-14: Fig. 1 - antenna is mis-spelled in the caption and the figure. The order of magnitude of the magnetic field should be given. It is not clear where the current sourced by voltage

source V_a is flowing, since there is no ground closing the circuit. It is not clear where voltage V_d is referenced, since there is only one line going into the amplifier. Fig c) suggests that the authors are sourcing a current across the constriction while the methods section states that they applied a voltage and merely measured the current. It is not clear how the squared quantity ΔI^2 can drop below zero in 1d.

A2-14: The spelling has been corrected. In the left hand side of the resonator, the electrodes are formed on the mesa which sustain the 2DEG so that the electrode and the 2DEG form a capacitor. The 2DEG is grounded. Hence V_a is applied across this capacitor, and V_d is measured with reference to the ground. To clarify this further the circuit symbols in Fig. 1a [and also Fig. S1] have been modified. Additionally the transverse magnetic field B_x applied by the antenna gate is of the order of 0.8 mT estimated from the input RF power. Finally Fig. 1d is ΔI not ΔI^2 and this has been corrected.

Q2-15: Fig. 2. Why does the oscillation amplitude rise with detuning rather than dropping from a maximum at zero detuning? Is there a reason why the signal ratio differs between the experiment and the theory?

A2-15: Because of the cubic (Duffing) nonlinearity in the harmonic potential of the mechanical resonator at strong driving, its motional spectrum shows a sawtooth-shape with the amplitude increasing as it is detuned from the fundamental mode frequency. The mechanical amplitude was adjusted in this way, instead of changing the driving voltage V_a , as a large V_a could lead to the system temperature increasing. To avoid such unwanted heating the V_a value is kept constant and the amplitude is adjusted via frequency detuning. In the theoretical modelling only a single spin coupled to the resonator is considered whilst the experimental NMR spectra is acquired from an ensemble of nuclei. Consequently ensemble averaging results in a signal ratio that differs from the theory.

Typos:

p.2 "exceeds" should be "exceed"

p. 3 "from GaAs/AlGaAs heterowafer" - there is an "a" missing

These typos have been corrected.

Reviewer #3

Q3:” This is an impressive experiment that I am not at all surprised was carried out at NTT. I commend the authors for having undertaken that experiment, and brought the electrically detected NMR to that level, i.e. to combine the NMR with dynamical strain. I have seriously thought in the past about making this experiment myself, so when I was asked to be the reviewer I was at first skeptical regarding the data and the results claimed. Proper reading of the manuscript by my student and I proved the work to be clear, systematic, well-engineered, and its interpretation well thought out. A non-specialist in the field could criticize the line shapes shown and could find them departing from what we expect from “traditional” NMR. But here, the signal-to-noise is sufficient, and the effect of the dynamical strain appears clear to my eye. The simulation also rings to me as correct. The data, methods and analysis are all complete, coherent and most importantly new. The technique developed and described in the paper is similar in reasoning to other ones that require nuclear spins of long relaxation time, however it goes further by utilizing the dynamic strain induced manipulation of the quadrupole interaction. Finally, this is a somewhat across the field paper that fits extremely well within the nature comm format and its mission. It bridges different aspect of physics, and also different communities. It is clear to me that it can be published in Nature Comm. The only caveat is that I am not yet convinced that this paper will become highly cited rapidly and this is because it is somewhat a “heroic effort” that will take time for others to catch up. This being said, I strongly feel this should be published in Nature Comm with any further delay.

I have some points below (some more important than others) that I wish the authors will consider before going to print. They are not obliged to agree with them, but as a friendly competitor in the field, I hope to receive a response from the authors where due diligence has been done.”

A3: The authors thank the referee for the comments and encouraging our work.

Following reviewer #3's comment, we carefully revised the manuscript as detailed below.

Q3-1: On page 3, the authors discuss the motional piezo voltage measured for different actuation voltages, and refer to figures 1b and 2a, although 2a does not seem relevant for this information.

A3-1: In Fig. 2a, the mechanical motion is driven with fixed actuation voltage $V_a = 40$ mV

instead of varying V_a to change the mechanical amplitude. Hence Fig. 2a has no information of actuation voltages. To avoid unwanted heating the V_a value was kept constant and instead the mechanical amplitude was adjusted by varying the frequency of the actuation. Consequently this figure shows the frequency detunings necessary to achieve the desired mechanical displacements.

Q3-2: On page 10, when developing the perturbation theory terms, the authors never mention why they neglect the real or imaginary parts in certain calculations; in fact Eq. 4 completely omits the imaginary part of the quadrupole operator whereas Eq. 6 omits the real part, without any explanation or justification. When looking at the diagonal and off-diagonal terms in the matrix of the Supp. Info, it becomes a bit clearer, however this should be made explicit to the readers.

A3-2: The real part of the quadrupole operator is a diagonal matrix, while the imaginary part is an off-diagonal matrix. Equation 4 is the first order in which only the diagonal part (i.e. real part) of the operator contributes to the energy shift so the imaginary part is omitted. Equation 6 is the second order in which only the off-diagonal part (i.e. imaginary part) of the operator contributes to the energy shift so the real part is omitted. In the revised manuscript, this point has been addressed in the Methods section F.

Q3-3: There is no discussion or characterization of errors of any kind. This seems very unfortunate; when developing a new technique such limitations should be addressed to a certain degree, especially if they are to be considered negligible at all (which I suppose is the case seeing as how they are not even visible in the plots).

A3-3: The NMR detection via the electronic transport measurements typically has a fluctuation of about 0.3 nA which originates from charge fluctuation in the heterostructure. Because of this error in transport measurements, the resulting signal-to-noise ratio is reduced to 2~3 as shown in Fig. 2d. In future work effort will be made to minimize this error by improving the layer structure and the nano-fabrication process to reduce these stochastic charge fluctuations. In the revised manuscript, we addressed these points in Methods.

A3-4: In some sentences, the use of parentheses for "(typically > MHz)" seems unnecessary, though I find the way this part was written a little off-putting.

Q3-4: The parentheses have been omitted from these sentences in the revised manuscript.

Q3-5: On numerous occasions the authors refer the reader to the "Method" section by writing "see Method". This should be written as "Methods" or "Method section".

A3-5: "Method" has now been re-written as "Methods"

Q3-6: Figure 2d shows two separate curves with different color data points, referred to as "simulation" and "experiment" by little lines of corresponding color; this seems inefficient and would be better with a standard legend.

A3-6: A standard legend has now been introduced to these figures.

Q3-7: The second paragraph on page 5 starts with a run-on sentence; a comma needs to be put somewhere or it needs to be split into two different sentences to be clear.

A3-7: In the revised manuscript care has been taken to eliminate such run-on sentences.

Q3-8: The curves in blue and orange of figures 4c-d with "x3" indicating the amplitude was multiplied by three (to be more visible to the reader) is odd; a second y-axis would suffice (and still no uncertainty values are visible!).

A3-8: A second y-axis on the right hand side has now been introduced to address this.

Q3-9: The use of "On the other hand" towards the end of the second paragraph on page 5 is out of place and does not follow with a subject contradicting the logic of the phrase preceding it.

A3-9: "On the other hand" has now been omitted.

Q3-10: The phrasing "For more advanced dynamic quantum manipulation of nuclear spins via electromechanical phonons requires stronger coupling between them" needs some work and refining.

A3-10: In the discussion section of the revised manuscript, possible experiments in which strong coupling could be achieved are considered by ensemble enhancement of the effective coupling strength needing 10^{10} nuclear spins where this number is accessible in

realistic experiments.

Q3-11: In Figure 1, "Mechanical motion" is not capitalized at all, let alone the same way as the other titles in the figures. The same goes for a couple of subtitles in figure 3.

A3-11: The subtitles are now capitalized.

Typos:

Page 7, paragraph 2: "these spin flip-flop events".

Page 7, paragraph 3: "Figs. 1d and 2b." (No comma after 2b)

Page 9, first paragraph: "Figs. S3a and S3b" (note: this occurs twice in the same paragraph).

Page 9, first paragraph: " $x_1 = 0$ " (no comma after the "0").

Page 9, paragraph 2: "the oscillating components (...) have no contribution in the first

These typos have been corrected.

Reviewers' Comments:

Reviewer #1 (Remarks to the Author):

The revision lead to a significantly improved manuscript, which can be recommended for publication in Nature Communications with some improvements suggested below.

In my opinion, the authors have fully addressed the comments of the reviewers. The only exception is question “A1-7”. Perhaps the authors misunderstood this question because it was not formulated clearly enough, so I should reformulate it. It is clear that the experiments presented here sample only the interaction of the mechanical mode with a small ensemble of nuclei that are polarized. In the context of the present work, this raises no objections. My question was rather about the potential consequences for any future schemes that seek to exploit this coupling between the nuclear spins and the phonons. My main comment is that the system studied here is intrinsically different from a textbook case of an “atom coupled to a cavity”. Instead it is akin to an observable atom (polarized nuclei here) coupled to a cavity (mechanical mode), which is also coupled to a large unobservable bath of atoms (nuclei that are not polarized). Thus while “atom coupled to a cavity” is essentially a closed system, the system studied here is an open system. Intuitively, one would expect that the large dormant bath of unpolarized nuclei would be a source of decoherence, disrupting the correlations between the phonons and the observable nuclei: potentially this decoherence is inherently so strong that it would never be possible to achieve the regimes required for QED in a system of nuclear spins coupled to a mechanical resonator. This issue appears to be inherent, since it is difficult to envisage a realistic system where all nuclei affected by the resonator mode strain could be uniformly polarized, manipulated and probed. I think it is an important point related to potential implications of this work. Could the authors comment on this issue in the “Discussion” section – ideally give an estimate of the effect of the dormant unpolarized nuclei on the coherent phenomena related to the observable nuclei.

Some further comments:

Page 12. Equation (5) in Methods, second term contains a $\langle 2/2 |$ state – typo? Moreover, the 2nd, 3rd, and 4th terms of this equation are explained, but what is the physical meaning of the first term?

Supplementary Fig. S3 a and b, the vertical scale is in seconds – is this correct? Please clarify what determines the time-period of the oscillations in these figures? Few-second long timescales seem to be rather long compared to the nuclear T2 (milliseconds) or T2* (tens of microseconds) in GaAs, one would expect that all oscillations decay within T2 or T2*? This needs some

explanation.

“...which indicates that the strain-mediated coupling observed in this system is weak” - would it not be more accurate to say “...which indicates that the strain-mediated coupling observed in this system corresponds to a weak coupling regime”?

Reviewer #2 (Remarks to the Author):

The authors have carefully addressed all of my concerns. I do not have any further objections to publication of the manuscript.

Reviewer #3 (Remarks to the Author):

I have read in details the rebuttal by the authors to all referees. In short, the new and improved manuscript greatly improves clarity of the paper. Notably, the addition of small details such as explaining the perturbation theory terms, providing explicit values of inherent errors of the system, fixing some minor typos and modifying the presentation of certain graphs and data in order to improve the overall clarity of the paper is appreciated and truly reflects the effort put into this work.

Perhaps more importantly was the reply of the authors to referee 1. While by all mean most of the referee's questions were sound and well taken, they have been extremely well adressed by the authors. Where my opinion greatly diverges is in the « taste and take » on the advance being made here. I quote here verbatim Ref 1 ,

« My overall opinion is that a considerable effort would be needed to make this work suitable for Nature Communication in terms of novelty of the results » ,

and frankly I do firmly believe the authors have upheld extremely high standard in the design of the device, the execution of the experiment, in the calculations made, and in the end the interpretation of their data. While I was dubious at first the I received the draft, I shall say that I was impressed by this well-executed experiment and as such, I view it as an important advance in a field, and within multi-cross-field. In my opinion, it is perhaps a new route being opened, a paradigm shift, or simply a technical « advance » *but* mostly likely it is a linear combination of the three that I hope nature communications will recognize by way of publication.

Point-by-point response to the reviewer 1.

Reviewer #1 (Remarks to the Author):

Q1-1: The revision lead to a significantly improved manuscript, which can be recommended for publication in Nature Communications with some improvements suggested below.

In my opinion, the authors have fully addressed the comments of the reviewers. The only exception is question “A1-7”. Perhaps the authors misunderstood this question because it was not formulated clearly enough, so I should reformulate it. It is clear that the experiments presented here sample only the interaction of the mechanical mode with a small ensemble of nuclei that are polarized. In the context of the present work, this raises no objections. My question was rather about the potential consequences for any future schemes that seek to exploit this coupling between the nuclear spins and the phonons. My main comment is that the system studied here is intrinsically different from a textbook case of an “atom coupled to a cavity”. Instead it is akin to an observable atom (polarized nuclei here) coupled to a cavity (mechanical mode), which is also coupled to a large unobservable bath of atoms (nuclei that are not polarized). Thus while “atom coupled to a cavity” is essentially a closed system, the system studied here is an open system. Intuitively, one would expect that the large dormant bath of unpolarized nuclei would be a source of decoherence, disrupting the correlations between the phonons and the observable nuclei: potentially this decoherence is inherently so strong that it would never be possible to achieve the regimes required for QED in a system of nuclear spins coupled to a mechanical resonator. This issue appears to be inherent, since it is difficult to envisage a realistic system where all nuclei affected by the resonator mode strain could be uniformly polarized, manipulated and probed. I think it is an important point related to potential implications of this work. Could the authors comment on this issue in the “Discussion” section – ideally give an estimate of the effect of the dormant unpolarized nuclei on the coherent phenomena related to the observable nuclei.

Response: The authors thank the reviewer for their fruitful comments on the manuscript.

To address this point the following sentences have been added to the discussion part:

“For application to a quantum coherent hybrid system, the decoherence induced by the unpolarized nuclear spins, which are coupled to mechanical motion at the clamping points (but unpolarized by the electron transport) needs to be suppressed. An optimised device structure where the mechanically strained clamping points completely overlap with the

nuclei polarised from the electron transport can in principle achieve this objective. Ultimately optical access to the sample will enable the nuclear bath in the entire resonator to be polarised thus enabling quantum coherent coupling between nuclei and phonons.”

Q1-2: Page 12. Equation (5) in Methods, second term contains a $|2/2\rangle$ state – typo? Moreover, the 2nd, 3rd, and 4th terms of this equation are explained, but what is the physical meaning of the first term?

Response: $|2/2\rangle$ was a typo and this should read $|3/2\rangle$. The first two terms describe the transition between $|1/2\rangle$ and $|3/2\rangle$ states, and the last two terms describe the transition between $|-1/2\rangle$ and $|-3/2\rangle$ states. To make this point clear in the revised manuscript the following sentence is now included: “where the first (last) two terms describe the transition between $|\text{ket}\{1/2\}$ and $|\text{ket}\{3/2\}$ ($|\text{ket}\{-1/2\}$ and $|\text{ket}\{-3/2\}$) states.”

Q1-3: Supplementary Fig. S3 a and b, the vertical scale is in seconds – is this correct? Please clarify what determines the time-period of the oscillations in these figures? Few-second long timescales seem to be rather long compared to the nuclear T2 (milliseconds) or T2* (tens of microseconds) in GaAs, one would expect that all oscillations decay within T2 or T2*? This needs some explanation.

Response: This was a typo and should read ms (milliseconds).

Q1-4: “...which indicates that the strain-mediated coupling observed in this system is weak” - would it not be more accurate to say “...which indicates that the strain-mediated coupling observed in this system corresponds to a weak coupling regime”?

Response: This sentence is now modified to: “which indicates that the strain-mediated coupling observed in this system corresponds to a weak coupling regime”.